# Serum Ketone Levels May Correspond with Preoperative Body Weight Loss in Patients Undergoing Elective Surgery: A Single-Center, Prospective, Observational Feasibility Study

**DOI:** 10.3390/ijerph19116573

**Published:** 2022-05-27

**Authors:** Paweł Kutnik, Michał Borys, Grzegorz Buszewicz, Paweł Piwowarczyk, Marcin Osak, Grzegorz Teresiński, Mirosław Czuczwar

**Affiliations:** 1II Department of Anaesthesiology and Intensive Care, Medical University of Lublin, Staszica 16, 20-081 Lublin, Poland; michalborys1@gmail.com (M.B.); piwowarczyk.pawel@gmail.com (P.P.); czuczwarm@gmail.com (M.C.); 2Laboratory of Forensic Toxicology, Department of Forensic Medicine, Medical University of Lublin, Jaczewskiego 8b, 20-090 Lublin, Poland; g.buszewicz@umlub.pl (G.B.); marcinosak@umlub.pl (M.O.); 3Department of Forensic Medicine, Medical University of Lublin, Jaczewskiego 8b, 20-090 Lublin, Poland; grzegorzteresinski@umlub.pl

**Keywords:** nutrition, prehabilitation, surgery, perioperative period, presurgical evaluation

## Abstract

Although nutritional-risk scoring systems allow the determination of the patient’s malnutrition at admission, additional tools might be useful in some clinical scenarios. Previous medical history could be unavailable in unconscious or demented patients. This study aimed to assess the ketone bodies serum levels in patients undergoing elective surgeries and to determine the possible correlation between ketone bodies serum levels and preoperative body weight loss. The study included 21 patients who underwent elective surgery. Hyperketonemia, defined as ketone bodies over 1 mmol/L, was observed in seven (33.3%) patients. Patients with hyperketonemia had significantly higher preoperative percentage body weight loss (*p* = 0.04) and higher nutritional risk scores prior to surgery (*p* = 0.04). Serum glucose and the percentage of body weight loss were associated with increased ketone bodies serum levels (Odds Ratios: 0.978 (0.961–0.996, *p* = 0.019) and 1.222 (1.0–1.492, *p* = 0.05), respectively). A significant linear correlation was found between the percentage of body weight loss and both ketones bodies (r^2^ = 0.25, *p* = 0.02) and albumin serum levels (r^2^ = 0.19, *p* = 0.04). Our study’s results might suggest a positive association between preoperative body weight loss and ketone bodies serum levels. The observation between increased ketone bodies serum levels in patients and preoperative body mass loss merits further research.

## 1. Introduction

The risk of malnutrition in inpatients remains very high and may range from 19% to 60% [1]. Furthermore, malnutrition prior to surgery is associated with a higher risk of perioperative complications and worse postoperative outcomes [2]. Disease-related malnutrition prolongs hospital stay and increases the costs of hospitalization [3]. The identification of patients at risk of malnutrition plays an important role in presurgical preparation, especially in cases that require major surgeries. Multiple preoperative assessment scales have been proposed to identify patients at risk of malnutrition, including the perioperative nutrition screen tool (PONS) and the nutritional risk score (NRS) [4,5]. Although screening tools play a major role in clinical practice, their sensitivity remains relatively low (62%) [6]. According to a study by Kyle et al., NRS-2002, nutritional risk index (NRI) and malnutrition universal screening tool (MUST) gave different sensitivity and specificity in comparison to the Subjective Global Assessment (SGA). As the majority of screening tools were validated by comparing varying combinations of other tools, often in different settings and populations, the results of the screening could sometimes provide confusing results [7]. Additionally, an impaired nutritional state and preoperative fasting might lead to an energy deficiency during surgery. Ketone bodies are produced as a result of an energy deficiency to support energy delivery to the cardiovascular, muscular, and central nervous systems [8]. Factors that impact the production of ketone bodies and, therefore, potentially predict energy deficiency during surgery, such as hypoglycemia, are of interest in many fields of medicine and are constantly being researched [9].

Since the introduction of the Global Leadership Initiative on Malnutrition (GLIM) criteria for the diagnosis of malnutrition, we received additional tools for nutritional evaluation. Non-volitional body weight loss became one of the phenotypic criteria for detecting and assessing the severity of malnutrition. In order to diagnose malnutrition, the patient needs to meet one phenotypic and one etiological criterion, and often the condition that meets the etiologic criterion is the indication for the surgery.

Introducing the nutritional therapy before surgery in patients with nutritional risk plays a major role in proper preoperative management, according to the European Society of Clinical Nutrition and Metabolism (ESPEN) [10]. Nevertheless, the majority of tools to determine potential malnutrition are often based on medical history and clinical judgment. The GLIM criteria introduced the objective validated methods to assess body composition as one of the phenotypic criterion options, such as dual-energy absorptiometry or bioelectrical impedance analysis, yet they are often unavailable in many hospitals.

Appropriate medical history and evaluation at admission usually allow the determination of preoperative body weight loss and NRS. However, in some cases, including unconscious or demented patients, thorough medical history is not always available to obtain. Thus, the additional tools might be useful in everyday clinical practice. To our knowledge, there has been no study aiming at measuring the presurgical levels of ketone bodies and evaluating the correlation between patients’ oncological status or presurgical nutritional status, such as body mass loss prior to surgery and ketone bodies serum levels.

Therefore, the primary aim of this study was to assess the serum levels of ketone bodies in patients undergoing elective surgeries as a surrogate for patients’ malnutrition in a tertiary teaching hospital. The secondary aim was to determine the possible correlation between the presurgical nutritional state and the levels of ketone bodies.

## 2. Materials and Methods

### 2.1. Research Materials

This was a prospective, non-interventional study performed at a tertiary university hospital. The study was approved by the Institutional Review Board of the Medical University (KE-0254/350/2018). Each patient signed the consent form before his/her inclusion in the trial. The study included 21 adult patients who were scheduled to undergo elective abdominal surgery associated with gastrointestinal diseases in a general surgery department. Patients with diabetes, liver diseases, known alcohol addiction, or thyroid disorders; those receiving steroid hormone drugs; or those following the keto diet were not included in the study. Moreover, patients with advanced stages of cancer, signs of nodular involvement, or metastasis were excluded from the study. The collected data included patients’ age; weight; body mass index (BMI); diagnosis at admission; planned surgery; oncological status; body mass loss prior to surgery; NRS, albumin, and protein serum levels, arterial blood results; and the number of hours of preoperative fasting.

Patients’ anthropometric and laboratory data were collected at a routine preanesthetic visit the day before surgery, during which the NRS and PONS scores were calculated. The number of hours of preoperative fasting was calculated based on the time of the patient’s last meal prior to surgery and the time of the start of the operation.

Preoperative body loss weight was defined as any weight lost up to six months before the surgery according to information obtained from the patient. The patient’s body mass loss, if any, was also divided by the patient’s primary weight to establish the percentage of weight loss prior to the surgery.

Before the induction of general anesthesia, a blood sample (2 mL) was collected from each patient to determine his/her ketone body levels. After the induction of general anesthesia but before the first incision, an arterial blood sample was collected to determine acid-base balance, glucose, and electrolyte serum levels. None of the enrolled patients received preoperative oral supplements, preoperative nutritional care, intravenous glucose solutions, or preoperative carbohydrate drinks before surgery.

### 2.2. Ketone Bodies Determination

To determine the ketone body serum levels, the blood samples were centrifuged within 15 min of collection at 3000 rpm for 10 min. The sample tube with the serum was then immediately cryopreserved in a −80-degree Celsius freezer and stored there until transportation to the laboratory.

Acetone determination was performed by the GC/FID-headspace 2 columns method [11] on a gas chromatograph Trace GC Ultra (2 FID detectors and split/splitless injector) coupled with a TriPlus HS autosampler (Thermo Fisher Scientific^®^., Waltham, MA, USA). Compounds were separated by using two fused-silica capillary columns BAC-1 and BAC-2 (Restek, Bellefonte, PA, USA) 30 m, 0.32 mm ID. The carrier gas was helium at a flow rate of 15.0 mL/min. The splitless mode was used. The injector temperature was set at 160 °C and the column temperature to 40 °C (constant). The analyzed samples were incubated in an autosampler agitator at 60 °C for 6 min then 300 μL of the gas phase of each sample were injected into the GC injector with an autosampler’s gas-tight syringe (heated at 62 °C). Sample preparation: In 2 headspace glass vials (10 mL), 200 μL of the blood sample and 200 μL of internal standard (IS, tert-butanol, 1 g/L in water) were added. The vials were then sealed with an aluminum cap and a silicon/teflon septum and placed in a headspace vials rack.

β-hydroxybutyric acid (BHBA) determination was performed by TSQ 8000 Evo Triple Quadrupole GC-MS/MS (collision gas-argon) with electron ionization (EI source) and TriPlus RSH autosampler (Thermo Fisher Scientific^®^, Waltham, MA, USA). Compounds were separated by using a 30 m, 0.25 mm i.d., 0.25 μm film Rxi-5ms fused-silica capillary column (Restek, Bellefonte, PA, USA). The carrier gas was helium at a flow rate of 1.0 mL/min. The splitless mode was used. Injection volume of the extract: 1 μL. The injector temperature was set at 265 °C. The column temperature programs were: initial temperature of 55 °C, increase to 160 °C at 15 °C/min, then increase to the final temperature of 220 °C at 30 °C/min, hold for 0.8 min. The transfer line temperature and MS source temperature were 265 and 255 °C, respectively. To avoid the saturation of the MS detector, a solvent delay time of 0.7 min was used. Quantitative analysis was performed by using the multiple reaction monitoring (MRM) mode with the characteristic transition from precursor ion to product ions: BHBA-diTMS derivative (*m*/*z*: quantifier 233.2→147.2; qualifiers 233.2→73.2 and 233.2→191.2) and internal standard GHB-D6-diTMS derivative (*m*/*z*: quantifier 239.2→147.2; qualifier 239.2→73.2). The concentration of BHBA in the tested material was calculated from the calibration curve (R^2^= 0.999) in the range of 0.5–300 µg/mL (6 BHBA calibration points in 6% bovine serum). The concentration of the internal standard (GHB-d6) in the samples was 10 µg/mL.

Sample preparation: The test material was mixed on a shaker (Vortex type, Vortex Genie® 2 mixer, Sigma Aldrich, Saint Louis, MO, USA) and transferred in an amount of 100 mL into Eppendorf-type vials (2 mL, Sigma Aldrich, Saint Louis, MO, USA). To each sample, 10 μL of a 100 μg/mL GHB-d6 solution (internal standard) was added, mixed, and subjected to liquid–liquid extraction (LLE), using 200 μL acetonitrile. Precipitation was carried out (over 30 s) by the gradual dosing of acetonitrile into the agitated sample. The organic phase was separated by centrifugation: 10 min, 10 °C, at 15,000 rpm. An amount of 250 μL of the supernatant was collected from above the pellet, transferred to the Eppendorf-type vials, and evaporated under a stream of nitrogen at 45 °C using a rotary evaporator (RapidVap N2/48, Labconco, Kansas City, MO, USA). The dry residue was dissolved in 50 PL ethyl acetate and 50 PL derivatization reagent—Sylon BFT (Sigma Aldrich, Saint Louis, MO, USA) was added. After mixing, the samples were incubated in a thermoblock for 30 min at 60 °C. The solutions were centrifuged for 2 min (14,000 rpm) and transferred to target chromatographic vials for GC-MS/MS triple-quad analysis.

### 2.3. Outcome

The main outcome of the study was to measure the concentration of ketone body serum levels in patients undergoing elective gastrointestinal surgery. Moreover, factors that correlate with the concentration of ketone body serum levels were established. Patients were divided into two groups according to their ketone bodies serum levels: below and above 1.0 mmol/L. This cut-off value was established based on the results obtained by Burstal et al. [12].

### 2.4. Statistical Analysis

Statistic data were collected in Microsoft Excel (Microsoft, Redmond, WA, USA). Categorical variables were presented as numbers and frequencies and analyzed using the Fisher exact test. Continuous variables were tested for normal distribution using the Shapiro–Wilk test. Normally distributed continuous variables were presented as means and standard deviations of the mean and analyzed using Student’s *t*-test. Non-normally distributed variables were presented as medians and interquartile ranges (IQRs) and analyzed using the Mann–Whitney U test. A logistic regression model was created to analyze variables that had an impact on the increased ketonemia. The variables in the models were presented as odds ratios (ORs) with 95% confidence intervals (CIs). All statistical calculations were performed using Statistica 13.3 (StatSoft Inc., Tulsa, OK, USA).

## 3. Results

In total, 21 subsequent patients, who consented to participate in this study, scheduled for different types of general surgery procedures, were recruited to the study. The study population included 10 patients with gastrointestinal tract cancer, including one with esophageal cancer, three with gastric cancer, five with colon cancer, and one with pancreatic cancer. Moreover, 11 patients with non-oncological gastrointestinal tract diseases were recruited, including six with hiatus hernia in the esophagus, three with cholecystic disease, and two with diverticulitis. None of the patients developed any severe postsurgical complications. Detailed descriptions of the patient demographics are presented in Table 1.

Table 1 presents the patients’ demographics. Study participants were divided into two groups according to measured serum ketones. Normally distributed variables are presented as means with standard deviations. The data statistics were calculated with *t*-tests. Non-normally distributed variables are presented as medians and interquartile ranges in parentheses, and probability was calculated with the Mann–Whitney U test. Categorical data are presented as numbers and percentages and were calculated using the Fisher exact test. NRS—nutritional risk score; BHB—beta-hydroxybutyrate.

Eight patients reported preoperative weight loss. The lowest reported preoperative weight loss was 6% and the highest was 15%. Seven patients were scheduled for surgery due to malignant diseases and one due to hernia hiatus esophagus meeting disease burden/reduce food intake etiological GLIM criteria. According to the above, all patients with reported preoperative weight loss met GLIM criteria for malnutrition. The lowest albumin serum level was 3.9 g/dL, and the highest was 4.8 g/dL. Patient characteristics based on existing preoperative weight loss are presented in Table 2.

Table 2 presents patient demographics. Study participants were divided into two groups according to preoperative body weight loss. Normally distributed variables are presented as means with standard deviations. The data statistics were calculated with *t*-tests. Non-normally distributed variables are presented as medians and interquartile ranges in parentheses, and probability was calculated with the Mann–Whitney U test. Categorical data are presented as numbers and percentages and were calculated using the Fisher exact test. NRS—nutritional risk score; BHB—beta-hydroxybutyrate.

### 3.1. Primary Outcome

Hyperketonemia, defined as ketone bodies serum levels above 1.0 mmol/L, was observed in seven (33.3%) patients. The highest concentration of ketone bodies serum levels found in the study was 11.64 mmol/L and the lowest was 0.1 mmol/L. Acidosis, defined as a pH below 7.3, was not detected in any of the participants.

### 3.2. Secondary Outcomes

Patients with hyperketonemia had significantly higher preoperative percentages of body weight loss (*p* = 0.04) and higher NRS prior to surgery (*p* = 0.04). Logistic regression models were calculated for the two groups (ketone bodies serum levels below and above 1.0 mmol/L, and patients with and without reported body weight loss). Two variables were associated with increased ketone bodies serum levels: serum glucose and percentage of body weight loss. The ORs for serum glucose and loss of body weight were 0.978 (0.961–0.996, *p* = 0.019) and 1.222 (1.0–1.492, *p* = 0.05), respectively. The area under the receiver operating characteristic curve (ROC) for this model was 0.785. NRS was associated with preoperative body weight loss in multivarious analysis; OR 8.71 (1.12–67.6, *p* = 0.04)

Additionally, the graphical representation of the association between any reported preoperative body weight loss and serum ketone levels is presented in Figure 1.

In our study, a significant linear correlation was found between the percentage of body weight loss and both ketone bodies and albumin serum levels. The Pearson’s correlation coefficient (r2) was 0.25, *p* = 0.02 for ketones and 0.19, *p* = 0.04 for albumin serum levels. Of the 21 patients, 10 were scheduled for surgery due to oncological conditions. No difference was found in the ketone bodies serum levels of oncological (1.01 [0.65–1.92] mmol/L) and non-oncological patients (0.52 [0.27–0.84], *p* = 0.13). The duration of preoperative fasting and the BMI, NRS, PONS, albumin, and protein serum levels prior to surgery did not correlate with the ketone bodies serum levels.

## 4. Discussion

The present results indicate a potential association between ketone bodies serum levels and preoperative body mass loss in patients undergoing elective gastrointestinal surgery. The main result of the study presents relatively high ketone bodies serum level elevation (33.3%) in patients undergoing elective surgery with reported preoperative body weight loss. It might provide a valuable clinical tool, which can be considered useful in unconscious and demented patients with unknown medical histories. The cut-off values of ketone bodies serum levels in non-diabetic adults have not been clearly defined. In our study, 33.3% of patients developed hyperketonemia, defined as ketone bodies serum levels over 1.0 mmol/L. This value was based on a study performed by Burstal et al. [12] In this study, hyperketonemia, defined as serum ketone bodies serum levels above 1 mmol/L, occurred in only 3% of patients (3/100). In comparison, in our study, 33.3% of patients (7/21) had ketone bodies serum levels over 1 mmol/L. The difference could be associated with different methods of measuring ketone bodies serum concentrations or different practices in preparing patients for surgery. However, our results suggest that the occurrence of hyperketonemia in the perioperative period might be higher than previously reported. Additionally, in the current study, no correlation was found between preoperative fasting time and ketone bodies serum levels. This outcome is consistent with the results of the study by Burstal, in which only a weak positive correlation between those two variables was observed. Hyperketonemia observed in patients was rather associated with the preoperative nutritional state than the inpatient fasting period.

The results of our study suggest that ketone body serum levels might be a more sensitive predictor of preoperative body weight loss in comparison to albumin serum levels. In the current study, both variables showed a significant correlation with the percentage of body weight loss; however, ketones body serum levels were more strongly associated with weight loss in our patients (Pearson’s correlation coefficient: ketones, *p* = 0.02 albumins, *p* = 0.04). As presented in Table 2, albumins levels and NRS score showed statistical significance along with ketone bodies serum levels over 1 mmol/L. Nevertheless, the lowest albumin level in our study population was 3.9 g/dL, which is above 3.5 g/dL, which is a cut-off value for hypoalbuminemia. In the case of NRS, almost half of the points are counted based on reported preoperative weight loss, thus the statistical significance of this result is self-explanatory. The higher representation of oncological patients in reported body weight loss could be associated with impaired digestion and absorption due to the malignant process [13]. A combination of multiple malnutrition detecting tools should be investigated in the future to determine the approach with the highest specificity and sensitivity of results.

According to the study results, two variables were associated with increased ketone bodies serum levels: serum glucose and percentage of body weight loss (OR 0.978 (0.961–0.996, *p* = 0.019) and 1.222 (1.0–1.492, *p* = 0.05), respectively. ROC = 0.785). We incorporated results obtained from 21 patients in a logistic regression model, which may raise doubts about a sufficient sample size. There is a rule indicating the need for at least 10 events for each degree of liberty of a multivariate analysis. However, according to the study by Forcino et al., a limited sample size does not always lead to a wrong estimation of the factors [14]. Thus, despite the small sample size of the current study, the results could still provide valuable data on factors associated with increased ketone bodies serum levels.

Considering the above results of the logistic regression models, serum glucose levels could be a predictor of increased ketone bodies serum levels. The decreased glucose levels in patients with reported body weight loss could be explained by reduced glycogen, promoting ketogenesis [15]. However, all of the patients from the study had glucose levels within the normal range, making it difficult to interpret in a clinical situation (over 70 mg/dL which is a cut-off value for hypoglycemia). Additionally, the ketone levels can be increased despite normal or high blood glucose levels in the case of low intracellular glucose levels, e.g., insulin resistance or diabetes, or upregulated by elevated thyroid hormones, cortisol levels, and catecholamines due to the increased breakdown of free fatty acids [16]. Thus, determining potential hyperketonemia by glucose level could be challenging and inaccurate in clinical practice.

There might be some rationale between preoperative body weight loss and increased ketone body serum levels. The unintentional body weight loss in some cases might lead to a loss of skeletal muscle mass and function. There are case report warnings about severe ketoacidosis in patients with chronically reduced muscle mass (due to spinal muscular atrophy) caused by decreased food intake [17]. The authors of this case report postulate that in patients with chronic malnutrition, due to glycogen deficiency even short-term starvation might lead to acute onset of ketoacidosis. In mild cases, we might observe hyperketonemia without acidosis.

In this study, perioperative serum glucose concentrations and presurgical body mass loss were found as factors associated with increased ketone bodies serum levels. In addition, a higher loss of body mass and lower glucose concentration were correlated with higher ketone bodies serum levels. Nevertheless, serum glucose levels do not always correspond with intracellular hypoglycemia, and this with ketogenesis, as serum glucose levels might not reflect insulin receptor functions, metabolic stress, or ensuing starvation leading to catabolic metabolism [18]. Thus, in some cases, ketone bodies serum levels might provide more accurate information about intracellular hypoglycemia, which is important from the clinical point of care. The prevention of hypoglycemia by intraoperative glucose infusion was confirmed to decrease ketone bodies serum levels [19]. Moreover, reports from the first trials investigating the impact of preoperative carbohydrate drinks showed that reducing perioperative hypoglycemia decreased ketone production [20]. However, without further investigating correlations between presurgical nutritional status and hypoglycemia, the knowledge on this topic remains elusive.

The clinical implications of existing hyperketonemia remain elusive. According to the literature, in patients with diabetic ketoacidosis, proper nutrition prior to the onset of acidosis reduces in-hospital mortality [21]. In that study, patients admitted to hospital with diabetic ketoacidosis were divided into two groups with or without protein-energy malnutrition. Patients with protein-energy malnutrition had an increased risk of developing sepsis, septic shock, acute kidney failure, acute respiratory failure, deep vein thrombosis, or pulmonary embolism. In our study, we did not evaluate the outcome of the surgery or postoperative period due to the heterogeneous population. However, malnutrition, and by that, possible unintentional body weight loss as one of the causes of malnutrition, is well established as a risk factor for potential complications. The above consideration was observed in diabetes mellitus patients, but hyperketonemia is sometimes observed in patients without a history of diabetes [22]. In this case report, poor nutrition prior to admission based on medical history was confirmed by the onset of refeeding syndrome during the hospitalization. Starvation ketoacidosis was also reported in healthy patients during the perioperative period, which due to acid-base disturbances can cause complications [23]. As presented above in the literature, hyperketonemia, or in severe cases ketoacidosis, might be a potentially serious clinical issue not only associated with diabetes mellitus but also an impaired nutritional state, affecting the patients’ outcomes.

This study had some limitations. This was a small, observational study with a heterogeneous population. Due to the nature of non-volitional weight loss, the data were obtained through the medical history obtained from the patients, which can be underestimated or overestimated. Although patients were screened for common metabolic disorders, we could not exclude potential asymptomatic rare metabolic disorders, such as mild l-carnitine deficiency, affecting ketone production and altering the result. Due to the heterogeneous population, a clear correlation between ketone bodies serum levels and patient outcomes was difficult to establish.

## 5. Conclusions

The results of our study might suggest that there is a positive association between ketone bodies serum levels and preoperative body weight loss. The observation between increased ketone bodies serum levels in patients and preoperative body mass loss merits further research. There is a need for studies in homogeneous populations to determine the impact of ketone bodies serum levels on patient outcomes.

## Figures and Tables

**Figure 1 ijerph-19-06573-f001:**
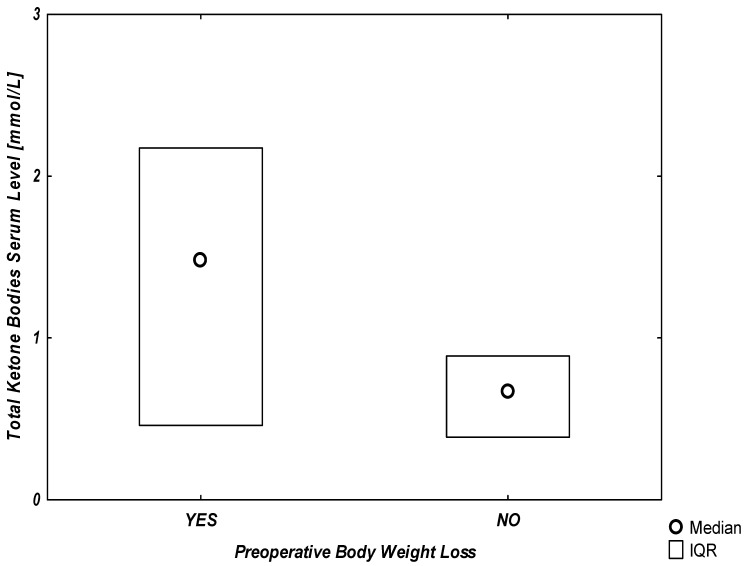
Median total ketone bodies serum levels divided in groups by preoperative body weight loss. Figure 1 Presents the difference between the median and interquartile range of total ketone bodies serum levels between the group with reported preoperative body weight loss and the group without any reports of weight loss. IQR—interquartile range.

**Table 1 ijerph-19-06573-t001:** Patient characteristics.

Variables	All Patients (*n* = 21)	Ketones below 1.0 mmol/L (*n* = 14)	Ketones over 1.0 mmol/L (*n* = 7)	*p*-Value
Age (y)	60.0 ± 12.0	57.6 ± 11.5	64.9 ± 12.6	0.73
Weight (kg)	83.52 ± 17.8	82.6 ± 17.5	81.9 ± 19.8	0.66
Height (cm)	168.4 ± 7.5	168.4 ± 8.3	168.4 ± 6.3	0.52
BMI (kg/m^2^)	29.28 ± 5.1	29.6 ± 5.0	28.6 ± 5.6	0.67
Females (%)	10 (47.6)	8 (60)	2 (16)	0.22
Preoperative fasting time (h)	19.75 [17.5–22.5]	19.9 [17.5–22.5]	19.5 [16.5–23.5]	1.0
Percentage of body mass loss (kg)	0 [0–10.45]	0 [0–0]	10.53 [0–14.75]	0.04
Oncological patient (%)	10 (47.6)	5 (40)	5 (66.7)	0.12
Glucose (mg/dL)	101 [89–108]	102 [99–108]	84.5 [73–107]	0.15
Albumins (g/dL)	4.3 [4.2–4.5]	4.4 [4.2–4.7]	4.3 [4–4.4]	0.29
Total Proteins (g/dL)	7.3 [7–7.7]	7.3 [7–7.7]	7.3 [6.5–7.4]	0.82
NRS	1 [0–2]	0.5 [0–1]	2 [0–3]	0.04
Acetone (mmol/L)	0.12 [0.06–0.23]	0.1 [0.04–0.12]	0.45 [0.23–0.84]	<0.001
BHB (mmol/L)	0.58 [0.35–1.18]	0.38 [0.2–0.58]	1.47 [1.18–1.93]	<0.001

**Table 2 ijerph-19-06573-t002:** Patient characteristics based on preoperative body weight loss.

Variables	All Patients (*n* = 21)	Patients without Reported Body Weight Loss (*n* = 13)	Patients with Reported Body Weight Loss (*n* = 8)	*p*-Value
Age (y)	60.0 ± 12.0	59.6 ± 12.4	60.6 ± 12.4	0.86
Weight (kg)	83.52 ± 17.8	88.4 ± 17.3	75.6 ± 16.7	0.11
Height (cm)	168.4 ± 7.5	170.4 ± 7.2	165.3 ± 7.4	0.13
BMI (kg/m^2^)	29.28 ± 5.1	30.4 ± 5.1	27.5 ± 4.8	0.22
Females (%)	10 (47.6)	5 (38.5)	5 (62.5)	0.28
Preoperative fasting time (h)	19.75 [17.5–22.5]	19.75 [19–22.5]	19.75 [15.75–22.75]	0.71
Ketones over 1 mmol/L	7 (33.3)	2 (15.3)	5 (62.5)	0.026
Oncological patient (%)	10 (47.6)	3 (23.1)	7 (66.7)	0.004
Glucose (mg/dL)	101 [89–108]	105 [100–126]	91.5 [73.5–101]	0.06
Albumins (g/dL)	4.3 [4.2–4.5]	4.5 [4.5–4.7]	4.2 [4–4.3]	0.02
Total Proteins (g/dL)	7.3 [7–7.7]	7.6 [6.7–7.8]	7.3 [6.5–7.4]	0.34
NRS	1 [0–2]	0 [0–1]	2 [1.5–3]	0.004
Acetone (mmol/L)	0.12 [0.06–0.23]	0.1 [0.06–0.13]	0.33 [0.08–0.67]	0.12
BHB (mmol/L)	0.58 [0.35–1.18]	0.56 [0.35–0.73]	1.12 [0.38–1.53]	0.23

## Data Availability

Some of the data can be obtained by contacting the corresponding author. Due to The DPR consent statement, some requests might be declined.

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
