# Peer review of "Serum Ketone Levels May Correspond with Preoperative Body Weight Loss in Patients Undergoing Elective Surgery: A Single-Center, Prospective, Observational Feasibility Study"

_ijerph, 2022, doi:10.3390/ijerph19116573_

Round 1

Reviewer 1 Report

The article is well restructured.

The bibliography does not follow the indications of the journal. Please restructure it.

Author Response

Thank you for noticing this issue. We changed the references according to the IERPH guidelines. However we are not sure about reference number 7 (lack of assigned pages numbers by publisher), number 10 (methodology description), 14 (epub) and 16 (chapter in internet book), those are not specified in the guidelines (we have changed it according to the IERPH guidelines for other types of articles)

Reviewer 2 Report

Thank you so much for your correction. I feel there is no concern. 

Author Response

Thank you very much for dedicating your time to helping us improve our manuscript. 

This manuscript is a resubmission of an earlier submission. The following is a list of the peer review reports and author responses from that submission.

Round 1

Reviewer 1 Report

They showed the measurement of preoperative serum ketone would be useful in terms of understanding the patient’s nutritional level correctly. It was so interesting for future research, but I have some comments.

  1. Their sample size was small to show their conclusion.
  2. To measure serum ketone level was valuable from economical point of view?
  3. Did they analyze the association preoperative factor and clinical outcome such as postoperative complication?
  4. What was the other influencing factor to understand the serum ketone level?

Reviewer 2 Report

In the introduction could the authors describe a little bit more the screaning tools and why their sensitivity is relatively low.

The present sutdy had established two aims:

  • the primary aim  was to assess the serum levels of ketone bodies in patients undergoing elective surgeries as a surrogate of patients’ malnutrition in a tertiary teaching hospital.
  • the secondary aim was to determine the possible correlation between the presurgical nutritional state and the levels of ketone bodies. 

In the discussion chapter the authors have to compare their results with more articles published on the same topic.